# Role of Wilting Time on the Chemical Composition, Biological Profile, and Fermentative Quality of Cereal and Legume Intercropping Silage

Cristiana Maduro Dias *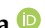, Hélder Nunes , Mariana Aguiar, Arnaldo Pereira, João Madruga and Alfredo Borba

Institute of Agricultural and Environmental Research and Technology, Faculty of Agrarian and Environmental Sciences, University of the Azores, Rua Capitão João d'Ávila, 9700–042 Angra do Heroísmo, Portugal; helder.pb.nunes@uac.pt (H.N.); 2021115445@uac.pt (M.A.); arnaldo.pereira16@hotmail.com (A.P.); joao.s.madruga@uac.pt (J.M.); alfredo.es.borba@uac.pt (A.B.)
* Correspondence: cristianarodrigues@gmail.com

**Abstract:** Agricultural production in the Azores primarily focuses on the livestock sector, notably, dairy production, where cows graze year-round in a rotational system. To maintain pasture productivity, farmers often rely on synthetic nitrogen fertilizers, which have adverse environmental impacts like ammonia emissions and nitrate leaching. Alternatively, nitrogen-fixing crops like legumes are explored as green manures to enhance soil quality and reduce dependence on chemical fertilizers. The traditional practice of using mixed forages of legumes and grasses, known as "outonos" or intercrops, has been crucial but is declining over time. These mixtures include plants such as lupins, Vicia faba, oats, and vetch, noted for their adaptability and nitrogen-fixing ability. Due to the high perishability of these crops, effective conservation strategies like ensiling are essential to preserve forage nutritional quality through controlled fermentation. This study evaluates the productivity and quality of intercrop forages in the Azores, focusing on fresh samples and silage prepared with wilting times of 0, 24, 48, and 96 h, followed by comprehensive chemical analyses. Results showed significant changes in fiber components (neutral detergent fiber, acid detergent fiber, and acid detergent lignin) with increased wilting time, leading to reduced digestibility. However, wilting improved dry matter content.

**Keywords:** green manure; legumes; consociation; sustainability; nitrogen fertilizers; feed conservation

## 1. Introduction

Optimizing animal feed is crucial in modern agriculture, not only due to its significant impact on production costs but also because of its direct influence on productivity and animal welfare. With over 50% of production costs linked to feed [1], it is essential to develop strategies that ensure efficient resource use while promoting sustainability in agricultural production [2].

In the Azores, agricultural production is dominated by the livestock sector, particularly the dairy industry. Cows graze on pastures year-round, with minimal stabling structures. To maintain pasture productivity for animal feed, farmers often apply synthetic nitrogen fertilizers to the soil to stimulate rapid plant growth, as nitrogen is a critical component of essential molecules. However, nitrogen fertilizers are also associated with environmental issues, such as ammonia and nitrogen oxide emissions and nitrate leaching into watercourses.

Research has shown that nitrogen deficiency is a significant limiting factor for many plant species in organic systems, underscoring the importance of nitrogen fertilization for high-demand crops [3]. Thus, incorporating nitrogen-fixing crops like legumes as commercial crops, cover crops, temporary green manures, or in intercropping systems, including agroforestry integration, is beneficial.

Legumes used as green manure can be incorporated into the soil, increasing its organic matter, improving water retention, and enhancing soil aeration [4]. Green manures improve soil's physical, chemical, and biological properties. They also help control weeds, prevent erosion, and enrich the soil with nitrogen, reducing the need for synthetic fertilizers [5].

Green manuring offers an economical and sustainable alternative to chemical fertilizers, which are often costly and environmentally harmful. The choice of green manure plants depends on local conditions, such as soil type, climate, subsequent crops, and soil management goals [6].

In the Azores, feeding cattle with "green manures" locally known as "outonos" was common for many years but has become rare today. "Outonos" typically consist of forage crops mixed with legumes and grasses, traditionally used in rotation with corn [7]. In the Azores, an intercrops "outonal" crop includes lupins, *Vicia faba* oats, and more recently, vetch.

*Lupinus luteus* is a resilient and nutritious plant known for thriving in acidic soils with low organic matter, contributing to soil fertility through nitrogen fixation [8]. *Vicia faba minor* varieties are recognized for their high productivity and protein content, adapting well to loamy soils unsuitable for many legumes, and can be grown as forage for green manure [9]. Sativa vetch (*Vicia sativa* L) is an annual legume planted in the fall, providing quality fodder and widely used in intercropping with cereals like triticale and oats. It adapts to various soils and climates and shows significant growth potential in spring [10]. Oats (*Avena sativa* L) are primarily grown for their grains and as animal fodder.

This study aims to evaluate both chemically and biologically the agricultural practice of "outonos", specifically the intercropping of *Lupinus luteus, Vicia faba minor* oats, and vetch, in their fresh form. Additionally, it investigates how wilting and ensiling affect the chemical composition of the silage. This research contributes to a deeper understanding of sustainable agricultural practices, aiming to optimize forage production for animal feed.

## 2. Materials and Methods

Different samples of "outonos" forage were collected on the island of Terceira, Azores (coordinates: 38°40′22″ N, 27°15′02″ W at an altitude of 103 m above sea level), to determine productivity, chemical composition, digestibility, and gas production.

### 2.1. Productivity Determination

To determine productivity, a 1 m$^2$ mold was placed in a previously defined area. All the vegetation inside the mold was then cut down and weighed using a hanging scale.

### 2.2. Sample Preparation

The samples were divided into two parts: one part fresh sample and the other for silage production, with different wilting times, as in Figure 1.

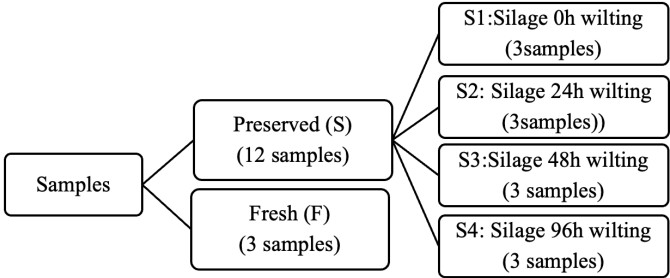

**Figure 1.** Scheme of the use and partition of the samples.

### Ensiling Procedure

Silage samples were prepared under controlled conditions to assess the effects of wilting duration on subsequent chemical and biological parameters. Samples were divided

into four treatments: S1: no wilting (0 h), S2: 24 h of wilting, S3: 48 h of wilting, and S4: 96 h of wilting. Each treatment was replicated in triplicate to ensure statistical validity and enable accurate evaluation of the results.

Initially, all samples were chopped into 2–3 cm pieces and thoroughly homogenized. Approximately 6 kg of each homogenized sample was weighed and packed into transparent mini silos (30 cm × 50 cm) made of a polyethylene-polyamide composite with a thickness of 0.14 mm. The samples were then compacted and vacuum-sealed to ensure anaerobic fermentation. The sealed silos did not allow for gases or effluents to escape and were stored in a dark place at ambient temperatures ranging from 20 to 25 °C in the Azores, for a period of 45 days. Each treatment was conducted in triplicate to ensure result reliability.

After the fermentation period, all samples underwent a rigorous drying process in a forced-air oven set at 65 °C until a constant weight was achieved. Post-drying, the samples were finely ground using a Retsch mill equipped with a 1 mm screen, ensuring uniformity and precision in preparation for subsequent analyses.

### 2.3. Chemical Analyses

For the analytical characterization, we employed established methods to determine several key parameters following standardized protocols. Dry matter (DM) content was determined using method 930.15, as outlined in the guidelines in [11]. Ash content was analyzed using the crude ash method (942.05), while ether extract (EE) content was assessed via method 920.39. Crude protein (CP) content was determined using the Kjeldahl method (method 954.01).

For fiber fractions, neutral detergent fiber (NDF), acid detergent fiber (ADF), and acid detergent lignin (ADL) were quantified using procedures detailed by Goering and Van Soest [12]. Hemicellulose (HEM) (1) and cellulose (CEL) (2) contents were estimated as follows:

$$HEM[\%DM] = NDF - ADF \tag{1}$$

$$CEL[\%DM] = ADF - ADL \tag{2}$$

### 2.4. Energy Estimates

Gross energy (GE) (3), digestible energy (DE) (4), metabolizable energy (ME) (5), and net energy for lactation (NEL) (6) were estimatesd according to the following equations [13–15].

$$GE[\frac{MJ}{KgDM}] = 18.45 - (0.88 \times NDF) \tag{3}$$

$$DE[\frac{MJ}{KgDM}] = GE \times (\frac{DMD}{100}) \tag{4}$$

$$ME[\frac{MJ}{KgDM}] = ED \times 0.82 \tag{5}$$

$$NEL[\frac{MJ}{KgDM}] = 0.101 \times GP + 0.051 \times CP + 0.11 \times EE \tag{6}$$

where *GP* is the gas production at 24 h (mL/200 mgDM), *NDF* is the neutral detergent fiber (% DM), *CP* is the crude protein (% DM), and *EE* is the ether extract (% DM).

### 2.5. Determination of Biological Parameters

In vitro dry matter digestibility (DMD) and in vitro organic matter digestibility (OMD) were assessed following the method described by Tilley and Terry [16], with modifications as outlined by Alexander and McGowan [17]. Gas production kinetics were quantified using the methodology proposed by Menke and Steingass [18].

The gas production kinetics were modeled using the equation developed by McDonald and Ørskov [19,20].

$$y = a + b\left(1 - e^{(-ct)}\right) \tag{7}$$

where in Equation (7), $y$ represents the cumulative gas production at time t, $a$ denotes the gas production from the immediately soluble fraction (mL per 200 mg of dry matter), $b$ indicates the gas production from the insoluble fraction (mL per 200 mg of dry matter), $c$ is the rate constant of gas production for the insoluble fraction (mL per hour), and $t$ refers to the incubation time in hours.

Rumen fluid utilized in each digestibility and gas production experiment was obtained following the protocols described by Dias et al. [21].

### 2.6. Statistical Analyses

Silages subjected to different treatments were evaluated: without resorting to wilting (0 h), and silage with the sample left to wilt for 24 h, 48 h and 96 h in order to determine whether there were statistically significant differences between them. All the data was statistically analyzed using SPSS Statistics Software v. 27 (IBM SPSS, Inc., Chicago, IL, USA).

The data obtained were submitted to the Shapiro–Wilk normality test, which indicated that the data followed a normal distribution,

The One-Way Analysis of Variance (One-Way ANOVA) was then used to identify whether there were significant differences between the means of the different silage treatments; when the $p$-value resulting from the ANOVA was less than 0.05, there were significant differences between the means of the treatments.

## 3. Results

### 3.1. Productivity

During the sampling process, productivity was measured, revealing a yield of 1948 kg of dry matter (DM) per hectare. This yield is based on samples collected from multiple locations. The mean DM yield was 1948 kg/ha, with a standard deviation (SD) of 150 kg/ha.

### 3.2. Chemical Analysis

Table 1 shows the forage quality parameters for fresh and conserved samples, including silage treated at different wilting times.

**Table 1.** Chemical composition of fresh and preserved "outonos" silage samples with different wilting times.

| Parameter | Fresh (F) | Preserved (Silage) | | | | SEM | *p*-Value |
|---|---|---|---|---|---|---|---|
| | | S1 | S2 | S3 | S4 | | |
| DM (%) | 10.00 [a] | 11.78 [b] | 13.13 [c] | 13.98 [c] | 26.47 [d] | 1.56 | <0.001 |
| CP (%DM) | 22.51 [a] | 21.97 [ab] | 21.66 [bc] | 20.77 [c] | 19.67 [d] | 0.28 | <0.001 |
| EE (%DM) | 1.96 | 2.05 | 2.05 | 2.07 | 2.08 | 0.02 | 0.443 |
| Ash (%DM) | 8.72 | 8.07 | 8.19 | 8.27 | 8.32 | 0.08 | 0.107 |
| IVDMD (%) | 77.31 [a] | 69.36 [b] | 65.29 [c] | 64.08 [c] | 62.36 [c] | 0.94 | <0.001 |
| IVOMD (%) | 68.3 [a] | 58.03 [b] | 56.95 [bc] | 55.63 [bd] | 48.12 [c] | 0.67 | <0.001 |

DM: dry matter, CP: crude protein, EE: ether extract, IVDMD: in vitro dry matter digestibility, IVOMD: in vitro organic matter digestibility, F: fresh sample, S1: silage 0 h wilting; S2: silage 24 h wilting; S3: silage 48 h wilting; S4: silage 96 h wilting; SEM: standard error of the mean. Different letters next to the respective value indicate significant differences in the nutritive parameters among sampling dates ($p < 0.05$). Means in rows bearing unlike superscripts differ at $p < 0.05$.

Dry matter (DM) increased significantly ($p < 0.001$) with wilting time, reaching its highest value in S4 (26.47%). This increase indicates that wilting promotes moisture loss in the forage. Crude protein (CP) decreased as the wilting time increased. Fresh forage had the highest CP content (22.51% DM), while S4 had the lowest (19.67% DM).

Neutral detergent fiber (NDF) and acid detergent fiber (ADF) contents increased with wilting time. NDF increased from 51.26% DM in fresh forage to 57.04% DM in S4, and ADF increased from 35.5% DM to 45.99% DM in the same period. Dry matter digestibility

(DMD) and organic matter digestibility (OMD) decreased significantly ($p < 0.001$) with wilting time, being lowest in S4. DMD decreased from 77.31% in fresh forage to 62.36% in S4, and OMD decreased from 68.3% to 48.12%. This reduction in digestibility is directly related to the increase in NDF and ADF contents.

There were no significant differences in ether extract (EE) or ash contents between treatments, indicating that these components remain relatively stable during the wilting process.

A detailed analysis of the changes in fiber components of the forage with increasing wilting times is provided in Table 2. There was a gradual increase in the values of NDF, from 51.26% DM in fresh forage (F) to 57.04% DM after 96 h of wilting (S4), indicating a higher total fiber content. Similarly, ADF increased from 35.50%DM in fresh forage (F) to 45.99% DM after 96 h of wilting (S4), suggesting an increase in less digestible components such as cellulose and lignin. ADL showed a significant increase from 5.39% DM in fresh forage (F) to 9.99% DM after 96 h (S4), highlighting increased resistance to digestion. In contrast, hemicellulose (HEM) decreased from 16.12% DM in fresh forage (F) to 9.80% DM after 24 h of wilting (S2), with a slight recovery in subsequent values, while cellulose (CEL) consistently increased from 30.11% DM in fresh forage (F) to 36.00% DM after 96 h of wilting (S4). These results indicate that prolonged wilting leads to an increase in less digestible fiber components, reducing the nutritional quality of the forage.

**Table 2.** Fiber composition of fresh forage and preserved "outonos" silage with different wilting times.

| Parameter | Fresh (F) | Preserved (Silage) | | | | SEM | *p*-Value |
| | | S1 | S2 | S3 | S4 | | |
|---|---|---|---|---|---|---|---|
| NDF (%DM) | 51.26 [ab] | 50.80 [a] | 53.06 [bc] | 54.79 [c] | 57.04 [d] | 0.67 | <0.001 |
| ADF (%DM) | 35.50 [a] | 39.75 [b] | 43.26 [c] | 44.54 [c] | 45.99 [c] | 1.03 | <0.001 |
| ADL (%DM) | 5.39 [a] | 7.37 [b] | 7.99 [bc] | 8.10 [c] | 9.99 [d] | 0.39 | <0.001 |
| HEM (%DM) | 16.12 [a] | 11.05 [b] | 9.80 [c] | 10.25 [b] | 11.00 [b] | 0.90 | <0.001 |
| CEL (%DM) | 30.11 [a] | 32.38 [b] | 35.27 [c] | 36.44 [c] | 36.00 [c] | 0.85 | <0.001 |

NDF: neutral detergent insoluble fiber; ADF: acid detergent insoluble fiber; ADL: acid detergent lignin; HEM: hemi- cellulose; CEL: cellulose; S1: silage 0 h wilting; S2: silage 24 h wilting; S3: silage 48 h wilting; S4: silage 96 h wilting; SEM: standard error of the mean. Different letters next to the respective value indicate significant differences in the nutritive parameters among sampling dates. $p < 0.05$ significant differences were found.

The quality parameters of "outonos" silage forage are presented in Table 3, specifically pH and the percentage of N-NH$_3$/N total, across different wilting times (S1, S2, S3, S4). The pH values of the silage decreased significantly ($p < 0.001$) with increased wilting time. The pH values were highest in S1 (4.61) and lowest in S4 (4.42). The percentage of ammonia nitrogen also showed significant differences ($p = 0.003$) across the different wilting times. The highest %N-NH$_3$/N total was found in S1 (13.25%) and the lowest in S4 (11.78%).

**Table 3.** Quality parameters of "outonos" silage forage.

| Parameter | Preserved (Silage) | | | | SEM | *p* Value |
| | S1 | S2 | S3 | S4 | | |
|---|---|---|---|---|---|---|
| pH | 4.61 [a] | 4.56 [ab] | 4.52 [b] | 4.42 [c] | 0.02 | <0.001 |
| %N-NH$_3$/N | 13.25 [a] | 12.94 [a] | 12.46 [bc] | 11.78 [c] | 0.18 | 0.003 |

S1: silage 0 h wilting; S2: silage 24 h wilting; S3: silage 48 h wilting; S4: silage 96 h wilting; SEM: standard error of the mean. Different letters next to the respective value indicate significant differences in the nutritive parameters among sampling dates. $p < 0.05$ significant differences were found.

The estimates of gross energy (GE), digestible energy (DE), metabolizable energy (ME), and net energy for lactation (NEL) of "outonos" forage in both fresh and preserved (silage) states are presented in Table 4. The data show that fresh forage (F) has higher values of GE, DE, and ME compared to preserved forage at different wilting times (0 h, 24 h, 48 h,

and 96 h). GE significantly decreased ($p = 0.03$) with increased wilting time, ranging from 13.98 MJ/KgDM at S1 to 13.43 MJ/KgDM at S4. Similarly, DE, ME, and NEL also decreased with wilting time, indicating that the wilting process negatively affects the energy value of the silage. However, NEL was slightly higher at S1 compared to fresh forage (F). The statistical analysis data indicated that there was no significant difference in NEL between samples S2, S3, and S4, and similarly, no difference was observed in NEL between the fresh (F) and S1 samples.

**Table 4.** Estimates of gross energy (GE), digestible energy (DE), metabolizable energy (ME), and estimated net lactation energy (NEL) of fresh and preserved "outonos" forage.

| Parameter | Fresh (F) | Preserved (Silage) | | | | SEM | *p*-Value |
|---|---|---|---|---|---|---|---|
| | | S1 | S2 | S3 | S4 | | |
| GE (MJ/KgDM) | 13.94 [a] | 13.98 [a] | 13.78 [b] | 13.63 [bc] | 13.43 [c] | 0.32 | 0.03 |
| DE (MJ/KgDM) | 10.78 [a] | 9.7 [ab] | 9 [b] | 8.73 [bc] | 8.38 [c] | 0.13 | <0.001 |
| ME (MJ/KgDM) | 8.84 [a] | 7.95 [b] | 7.38 [bc] | 7.16 [c] | 6.87 [c] | 0.18 | <0.001 |
| NEL (MJ/KgDM) | 6.18 [a] | 6.48 [a] | 5.67 [b] | 5.61 [b] | 5.51 [b] | 0.29 | 0.02 |

S1: silage 0 h wilting; S2: silage 24 h wilting; S3: silage 48 h wilting; S4: silage 96 h wilting; SEM: standard error of the mean. Different letters next to the respective value indicate significant differences in the nutritive parameters among sampling dates. $p < 0.05$ significant differences were found.

The cumulative fitted values of gas production of "outonos" forage in both fresh and preserved (silage) states over different incubation times (4, 8, 12, 24, 48, 72, and 96 h) are presented in Table 5. The data indicate that fresh forage had significantly higher gas production ($p < 0.001$) compared to preserved forage at all incubation times. As the wilting time for the silage increased (from S1 to S4), the gas production decreased.

**Table 5.** Cumulative fitted values of gas production (mL/200 mgDM) of fresh and preserved "outonos" forage.

| Incubation Time (h) | Fresh (F) | Preserved (Silage) | | | | SEM | *p*-Value |
|---|---|---|---|---|---|---|---|
| | | S1 | S2 | S3 | S4 | | |
| 4 | 4.82 [a] | 5.76 [b] | 4.49 [c] | 2.93 [d] | 3.40 [e] | 0.5 | <0.001 |
| 8 | 11.06 [a] | 13.07 [b] | 9.67 [c] | 8.46 [d] | 8.57 [e] | 0.49 | <0.001 |
| 12 | 16.55 [a] | 19.11 [b] | 14.09 [c] | 13.16 [d] | 13.01 [d] | 0.95 | <0.001 |
| 24 | 29.38 [a] | 31.65 [b] | 23.82 [c] | 23.44 [c] | 22.90 [d] | 1.06 | <0.001 |
| 48 | 44.11 [a] | 42.73 [b] | 33.63 [c] | 33.63 [c] | 33.11 [c] | 1.18 | <0.001 |
| 72 | 50.95 [a] | 46.27 [b] | 37.42 [c] | 37.47 [c] | 37.19 [c] | 1.46 | <0.001 |
| 96 | 54.14 [a] | 47.40 [b] | 38.88 [c] | 38.92 [c] | 38.82 [c] | 1.76 | <0.001 |

S1: silage 0 h wilting; S2: silage 24 h wilting; S3: silage 48 h wilting; S4: silage 96 h wilting; SEM: standard error of the mean. Different letters next to the respective value indicate significant differences. $p < 0.05$ significant differences were found.

The in vitro gas production kinetics parameters for both fresh and preserved "outonos" forage are shown in Table 6. Parameter a (mL/0.2 gDM) showed significant differences between all treatments ($p < 0.001$), with values ranging from –2.27 in fresh forage (F) to –3.58 in preserved silage (S4). Specifically, the fresh forage (F) had a higher value compared to S1 (–1.58), S2 (–2.63), S3 (–3.08), and S4 (–3.58).

**Table 6.** In vitro gas production kinetics parameters of fresh and preserved "outonos" forage.

| Kinetics Parameters | Fresh (F) | Preserved (Silage) | | | | SEM | *p*-Value |
|---|---|---|---|---|---|---|---|
| | | S1 | S2 | S3 | S4 | | |
| a (mL/0.2 gDM) | −2.27 [a] | −1.58 [b] | −2.63 [c] | −3.08 [d] | −3.58 [e] | 0.18 | <0.001 |
| b (mL/0.2 gDM) | 59.18 [a] | 51.02 [b] | 43.37 [c] | 42.54 [d] | 41.39 [e] | 1.8 | <0.001 |
| c (mL/h) | 0.0322 [a] | 0.047 [b] | 0.0406 [c] | 0.0396 [d] | 0.0382 [e] | 0.01 | <0.001 |
| tlag (h) | 1.20 [ab] | 0.97 [a] | 1.00 [ab] | 1.70 [c] | 2.10 [d] | 0.13 | 0.001 |

S1: silage 0 h wilting; S2: silage 24 h wilting; S3: silage 48 h wilting; S4: silage 96 h wilting; SEM: standard error of the mean. Different letters next to the respective value indicate significant differences. *p* < 0.05 significant differences were found.

Parameter b (mL/0.2gDM) also exhibited significant differences (*p* < 0.001), decreasing from 59.18 in fresh forage (F) to 41.39 in preserved silage (S4).

For parameter c (mL/h), significant differences were observed (*p* < 0.001), with values ranging from 0.0322 in fresh forage (F) to 0.0382 in preserved silage (S4).

The lag time (tlag) showed significant variation (*p* = 0.001). Fresh forage (F) had a shorter lag time compared to the preserved samples S3 (1.70 h) and S4 (2.10 h), and a similar lag time compared to S1 (0.97 h) and S2 (1.00 h).

## 4. Discussion

The silage process involves the fermentation of stored crops, such as corn, sorghum, or grasses, which are cut and compacted in silos to create an oxygen-free environment. Silage production is essential to meet the need for a constant and nutritious feed supply for livestock, ensuring continued milk and meat productivity throughout the year, regardless of climatic and seasonal variations. Additionally, silage is an efficient solution for utilizing forage surpluses, reducing waste, and contributing to the sustainability of agricultural activities [22,23].

To maintain pasture productivity, farmers often rely on synthetic nitrogen fertilizers, which are essential for rapid plant growth but are associated with environmental issues such as ammonia emissions and nitrate leaching [3]. On the other hand, nitrogen-fixing crops like lupine and vetch have proven effective when grown in combination, adapting well to various soil and climate conditions [24]. These crops (like lupine and vetch) can be used as green manure to improve soil quality, thereby reducing the need for chemical fertilizers [5,25].

The average productivity obtained for these crops was 1948 kg of dry matter per hectare, based on multiple samples collected at a time of year when the photoperiod is short and temperatures are lower, factors that generally reduce plant growth. Soil saturation due to excess rainfall, together with a decrease in temperature and photoperiod, significantly limits the production potential of pastures. However, the yields found in this study are higher than those reported by [26], who, in a perennial ryegrass pasture under temperate oceanic climate conditions, achieved an average yield of 65.4 kg of dry matter per hectare in one of the spring months. This contrast highlights the effectiveness of these crops, even in adverse environmental conditions.

Despite the high productivity potential, in the Azores, the practice of using mixed legume and grass forages, known as "outonos", has declined. This decline is clear even though it is effective when rotated with crops such as corn [7]. Traditionally, these mixtures include lupins, *Vicia faba*, oats, and vetch, plants recognized for their adaptability and nitrogen-fixing capability [8–10,27]. Therefore, promoting the cultivation of "outonos" could enhance forage productivity and sustainability in the region.

Although these crops are highly perishable due to their high moisture content, confirmed in this study by the dry matter (DM) value of fresh samples being only 10% (Table 1), effective preservation strategies such as ensiling are essential to maintain the nutritional quality of the forage through controlled fermentation [28]. Ensiling legumes like alfalfa (*Medicago sativa* L.) or red clover (*Trifolium pratense* L.) requires special care due to their

high protein content and low soluble carbohydrate content, which favors fermentation by clostridia [29]. Wilting can contribute to better fermentation of these legume silages [30]. The DM content is a critical parameter in silage production, directly influencing the fermentation process and the stability of the final product [31]. In this study, wilting significantly (<0.001) increased the dry matter (DM) content of the forage, especially after 96 h (S4) of wilting (26.47%). The increase in DM is beneficial because forage with higher DM content tends to ferment better, producing less effluent and increasing storage efficiency [32]. This increase is crucial because the reduction in moisture decreases the activity of undesirable microorganisms, such as clostridia, which can compromise the quality of the silage [33].

The pH of silage is a critical indicator of the efficiency of the fermentation process. Lower pH values are desirable because they indicate predominant lactic acid fermentation, contributing to the preservation of the forage [34]. In this study, a progressive decrease in pH was observed with increasing wilting time. A pH below 4.5, as observed in the conditions of 48 (S3) and 96 h (S4) of wilting, is ideal for inhibiting the growth of undesirable microorganisms [34]. These values are consistent with previous studies on ryegrass silage, where similar results were observed [35–37]. However, it is important to note that lower pH values alone do not necessarily reflect the dominance of lactic acid fermentation. Future studies should include the analysis of organic acids and alcohols to provide a more comprehensive understanding of the fermentation process.

The proportion of ammonia nitrogen reflects the proteolysis within the silo. Elevated values of ammonia nitrogen indicate the development of undesirable bacteria and can suggest that soluble nitrogen is not effectively assimilated by rumen bacteria, leading to protein value losses [38]. However, nitrogen can be utilized by rumen microorganisms if there are sufficient readily available carbohydrates in the diet.

For the samples considered in this study, the average $\%N\text{-}NH_3/N$ total values are within the expected range for grass silages, which can be considered of intermediate or good quality [39]. The reduction in ammonia nitrogen content with increasing wilting time suggests that wilting limits protein degradation, resulting in greater protein retention in the silage. This effect is particularly important for maintaining the nutritional value of the silage.

Wilting, by reducing the initial moisture content of the forage, contributes to achieving stability more quickly [40,41].

The data show that wilting increases dry matter (DM), thereby potentially reducing effluent production, which is beneficial for environmental conservation and waste management. However, this advantage comes with a trade-off: reduced feed value and lower energy concentration of the forage (Table 4). The gross energy (GE) decreased from 13.94 MJ/KgDM in fresh forage to 13.43 MJ/KgDM in silage S4 ($p$ = 0.03), indicating a loss of soluble energy components during wilting. Digestible energy (DE) also showed a significant decrease, from 10.78 MJ/KgDM in fresh forage to 8.38 MJ/KgDM in silage S4 ($p$ < 0.001), likely due to an increase in less digestible fibrous components. Similarly, metabolizable energy (ME) decreased from 8.84 MJ/KgDM in fresh forage to 6.87 MJ/KgDM in silage S4 ($p$ < 0.001), reflecting a lower efficiency in converting digestible energy into metabolizable energy. Net energy for lactation (NEL) also showed a notable decrease, ranging from 6.18 MJ/KgDM in fresh forage to 5.51 MJ/KgDM in silage S4 ($p$ = 0.02).The difference of 0.7 MJ NEL/KgDM between fresh forage and silage S4 resulted in a significant energy loss, considering a yield of almost 2 t of DM/ha, leading to a reduction of 1400 MJ NEL/ha in energy yield. While wilting can reduce effluent production, the associated energy losses due to reduced forage nutritional quality must be considered. Thus, the benefits of wilting in terms of reducing effluent and quickly stabilizing silage must be balanced against the disadvantages of reduced feed value and energy concentration.

In the samples from this study, it was observed that fresh forage had the highest crude protein content (22.51% DM), which decreased with increasing wilting time, reaching 19.67% in S4 (Table 1). These values are in line than those found by Hartinger et al., [42], who reported average CP contents for lucerne silages of between 4.41 and 24.88%DM

The variation in CP content was minimal among the different treatments, suggesting that wilting did not have a large impact on protein concentration.

NDF, ADF, and ADL are the chemical components most commonly used to predict forage digestibility [43]. Regarding preservation, in the case of grass silage, the values obtained indicate the amount of substrate available for fermentation. High values suggest overly mature grasses with fewer available free sugars, which may not ferment sufficiently to lower the pH to the ideal range for good preservation. In the case of corn silage, fiber values are not good indicators of the amount of available substrate, as corn silage generally has enough sugars to complete fermentation regardless of fiber content [44].

Significant changes ($p < 0.001$) in the fibrous components of the forage (Table 2) with increasing wilting time were evident, notably in NDF, ADF, and ADL. This indicates a higher total fiber content and less digestible components, suggesting reduced digestibility and nutritional value, as confirmed by decreases in dry matter digestibility (DMD) and organic matter digestibility (OMD) (Table 1). DMD and OMD values dropped significantly after 96 h of wilting, likely due to increased lignin content, which is highly resistant to digestion. The initial reduction in hemicellulose, followed by a slight increase, suggests an initial loss of fermentable carbohydrates, crucial for efficient fermentation. The increase in lignin content may further compromise the quality of the silage [45].

The reduction in forage digestibility and nutritional value due to increased wilting time is further supported by gas production data (Table 5). The amount of gas produced in in vitro fermentation reflects the extent of fermentation and the digestibility of forage [46], being directly proportional to the rate at which the substrate is degraded [47]. According to Chesson and Forsberg [48], during the initial incubation period, the soluble and rapidly fermentable fraction of the substrate (soluble carbohydrates) is fermented and microbial protein is synthesized. Once this phase is completed, the fermentation of insoluble but potentially degradable components, such as the NDF fraction, begins.

The fermentation kinetics, described using McDonald's methodology [19], suggest that longer wilting times reduce gas production from the immediately soluble fraction (a). Fresh forage consistently showed higher gas production across all incubation times, indicating better fermentability and higher digestibility. In contrast, silage samples, particularly those with longer wilting times, exhibited lower gas production values. For instance, at 96 h of incubation, fresh forage produced 54.14 mL of gas per 200 mg DM, whereas silage with 96 h of wilting (S4) produced only 38.82 mL. This trend of decreasing gas production with increasing wilting time reflects the increase in fiber content, especially cellulose and lignin, which are less fermentable, thus reducing the amount of carbohydrates available for fermentation.

## 5. Conclusions

This study demonstrated that using intercropped forages of legumes and cereals is an effective and sustainable practice for producing high-quality silage. Wilting proved to be an important method for increasing dry matter content despite the increase in fibrous components such as NDF, ADF, and ADL. However, prolonged wilting resulted in lower gas production, associated with higher cellulose and lignin content. Additionally, there was a reduction in dry matter digestibility (DMD) and organic matter digestibility (OMD), although the values remained acceptable for animal feed.

Future research should explore alternative or complementary wilting methods that preserve forage nutritional quality without increasing indigestible fibers. Furthermore, additional studies on the impact of different plant species and forage combinations on the ensiling process and nutritional quality would provide valuable insights for improving silage production practices.

**Author Contributions:** Conceptualization, C.M.D. and H.N; methodology, M.A.; investigation, A.P.; resources, J.M. and A.B.; data curation, H.N.; writing—original draft preparation, C.M.D.; writing—review and editing, H.N., J.M. and A.B.; supervision, A.B. All authors have read and agreed to the published version of the manuscript.

**Funding:** This work was funded by NOTS—NitroOrganic to Soils PRR-C05-i03-I-000020; Forest Research Centre (CEF), a research unit funded by Fundação para a Ciência e a Tecnologia (FCT), Portugal, grant number UIDB/00239/2020; the Laboratory for Sustainable Land Use and Ecosystem Services—TERRA (LA/P/0092/2020); and the FCT—Fundação para a Ciência e Tecnologia, through research grant 2020.06612.BD.

**Institutional Review Board Statement:** Not applicable.

**Data Availability Statement:** The data presented in this study are available on request from the corresponding author.

**Conflicts of Interest:** The authors declare no conflicts of interest.

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
