# Peer review of "Role of Wilting Time on the Chemical Composition, Biological Profile, and Fermentative Quality of Cereal and Legume Intercropping Silage"

_fermentation, doi:10.3390/fermentation10090448_

Round 1
Reviewer 1 Report
Comments and Suggestions for Authors The topic of the manuscript: „Role of Wilting Time on the Chemical Composition, Biological Profile, and Fermentative Quality of Cereal and Legume Intercropping Silage” falls within the thematic scope of FERMENTATION journal.The aim of the research was to determine the effect of treatment (wilting and ensiling) on ​​changes in the chemical composition and biological properties of intercropping Lupinus luteus, Vicia faba oat, and vetch. This research is very important due to the possibilities of introducing sustainable agricultural practices (e.g. optimization of feed production). Below is a list of comments and observations - in the attached pdf file I have included all comments to the text, including those that I do not mention here because they are minor.
The manuscript requires corrections listed below:
Note to the entire manuscript - please include the indicated methods of citation, including the names of the authors of the publication (for example line 117).
1. Abstract – lines: 22 - expand abbreviations; provide full names, not abbreviations,
2. Materials and methods - the methodology requires supplementing some information (for example lines: 94, 135-138),
3. Results –
a) description of some results is missing in the text - requires more detailed analysis and description of the results (for example lines 203-210),
b) there is no statistical analysis of the impact of time in individual variants S1 - S4 (Table 5) - this is a strong and important comment - the lack of this analysis makes it impossible to assess the impact of time on changes in individual S1-S4 variants
c) Figure 2 should be removed since it shows exactly the same results (even without statistical analysis) as those in Table 5,
4. Conclusions - The authors refer in the Conclusions to the conditions in the Azores, I think they should refer more generally to the conditions in different regions of the world, in Central Europe silage is also used to feed cows. Then the results obtained by the authors would be possible to use by researchers from other regions.
All comments were introduced in the review mode to the attached pdf file.

Author Response
Thank you very much for your thorough review and valuable feedback on our manuscript. We greatly appreciate the time and effort you invested in providing detailed comments and suggestions.
We have carefully reviewed all your suggestions and have made the necessary modifications to the manuscript accordingly. Your insights have significantly improved the quality and clarity of our work.
Reviewer 2 Report
Comments and Suggestions for Authors
Review
of the article entitled:
Role of Wilting Time on the Chemical Composition, Biological Profile, and Fermentative Quality of Cereal and Legume Intercropping Silage.
fermentation-3132695
General comments:
The paper deals with an important topic of sustainable dairy farming, under the specific conditions of the Azores with the aim at reducing synthetic N fertilizer use by legume intercropping.
The reviewer is not quite well familiar with the conditions of pasture use in the Azores, as are probably not many other readers of the journal. So, the background of the study needs better explanation. It seems that pastures on the island are not permanent otherwise there would not be land available for intercrops. Or are the pastures, supposedly based on ryegrasses, used for a few years and then intercrops are planted?
Unfortunately, major analytical parameters, including organic acids and alcohols, needed to evaluate fermentation quality, are not presented. Also, more information need to be provided in M&M section.
The paper as it stands is not suitable for publication. Major revision is required. Spelling and language need proper check.
Some specific comments
Abstract
The abstract seems to be more of an introduction and should be rephrases. Provide info on experimental design and the most important results.
M&M
L74: What does “different forage samples” mean. Describe experimental design. Were samples take from different locations? Give composition of outonos mix and seed rate/ha!
L78: was the plot for yield determination replicated?
L81: How many samples were taken from the field before division? Were they mixed before division or were analyses carried out on the samples taken from a given location?
L83: Why was no silage produced from fresh sample? Describe wilting procedure. In the field, in a barn…
L89/90: replications are simply needed for statistical evaluation and not to increase robustness and reduce variability. Rephrase.
L91-94: what bag? How compacted? What storage temperature? Why storage length of 45 days only? Too short to detect clostridia development as activity may occur after extended periods of storage.
L129-132: Hard to understand. Provide statistical model.
L137. What test for multiple comparisons among means was used? What is presented in Tables: means or Least Square means?
Results
Replace productivity by yield.
L141-142: Delete words like impressive. Was the presented DM yield based on one sample or samples from more locations? Present not only mean but also standard deviation (or SEM).
L145-146: Delete sentence “The table also….”, this is obvious.
L172-174: Rephrase. Example: Means (or LSMeans) in rows bearing unlike superscripts differ at p<0.05 (test?)
Table 2: check presentation of means. There are still values with , separation but it needs to be .
Line 199, Table: Was fresh forage not ensiled? If so, explain in M&M section why not.
%N-NH3/N: I think the dimension should be NH3-N in % of total N!
Were other important silage traits, e.g. concentration of lactic and other organics acids as well
as alcohols analyzed?
Discussion
This section is far too long, especially because it is, to a certain extent, a repetition of results.
L245-L253: too general, can be deleted.
L260-L267: If this statement is based on measurement of yield on just one sample, the whole section must be deleted as measurement not repeated.
L277-L280: Care should be taken in comparing outonos mixes with pure alfalfa and clover. They certainly behave differently. Reference to data on pure faba beans/vetch or oat/bean/vetch mixes more appropriate. Literature search required.
L289-295: Lower pH do not necessarily reflect the dominance of lactic acid fermentation. Therefore, presentation of acid and alcohol concentration is required.
Usually, pH increases with increasing DM level, but the authors see the opposite. It is not just the pH that inhibits the growth of undesired bacteria but pH in relation with the DM level (see Weissbach 2011, provided). The critical pH to inhibit clostridia was lower than required in all treatments except S4. So, treatments S1-S3 may have undergone clostridial fermentations by producing butyric acid and other higher homologues, either from sugar or from lactic acid, which could explain the higher pH in those treatments than in S4. Thus, to explain the data, analysis of organic acids must be done!
L296: Avoid “degradation of protein” and rather use proteolysis.
L297-305: Difficult to understand. NH3 can easily be used by rumen microorganisms given that there are enough easily available carbohydrates in the diet.
The data show that wilting increases DM thereby potentially reducing effluent production. Bit this comes with a price: reduced feed value, lower energy concentration. This needs to be properly discussed! Difference 0.7 MJ NEL/kg DM between fresh and S4!. Considering the yield of almost 2 t of DM/ha, this results in 1400 MJ NEL/ha lower energy yield. What would be the losses of energy by effluent production?
L306-311: Caution in comparing outonos with grass! Completely different material! Reference 42 did not use grass but lucerne! Check!
L319-356: In principal repetition of results! Delete and rephrase, respectively. .
Conclusions
This sections needs to be rephrased, and shortened, and conclusions be presented.
L363: too general. Depends on animal category, performance level and other components of the diet.
L365-L369: This was not tested in the study.
Comments on the Quality of English LanguagePls consult with language professional
Author Response
Thank you very much for your thorough review and valuable feedback on our manuscript. We greatly appreciate the time and effort you invested in providing detailed comments and suggestions.

Round 2
Reviewer 2 Report
Comments and Suggestions for Authors
Article is OK for publication now.